# Impact of the COVID-19 Lockdown in a European Regional Monitoring Network (Spain): Are We Free from Pollution Episodes?

**DOI:** 10.3390/ijerph182111042

**Published:** 2021-10-20

**Authors:** Gotzon Gangoiti, Maite de Blas, Maria Carmen Gómez, Ana Rodríguez-García, Eduardo Torre-Pascual, Estíbaliz García-Ruiz, Estibaliz Sáez de Cámara, Iñaki Zuazo, José Antonio García, Verónica Valdenebro

**Affiliations:** Faculty of Engineering, University of the Basque Country UPV/EHU, 48013 Bilbao, Spain; maite.deblas@ehu.eus (M.d.B.); mariacarmen.gomez@ehu.eus (M.C.G.); ana.rodriguezg@ehu.eus (A.R.-G.); eduardo.delatorre@ehu.eus (E.T.-P.); estibaliz.garcia@ehu.eus (E.G.-R.); estibaliz.saezdecamara@ehu.eus (E.S.d.C.); joseignacio.zuazo@ehu.eus (I.Z.); joseantonio.garciaf@ehu.eus (J.A.G.); veronica.valdenebro@ehu.eus (V.V.)

**Keywords:** air quality, COVID-19 lockdown, pollution episodes, ozone, PM_2.5_

## Abstract

The impact of the lockdown, during the period from March to June in 2020, upon the air quality of the Basque Country in northern Spain is analyzed. The evaluation accounts for the meteorology of the period. Daily and sub-daily analysis of aerosol and ozone records show that the territory was repeatedly affected by episodes of pollutants from outer regions. Three episodes of PM_10_ and ten of PM_2.5_ were caused by transported anthropogenic European sulfates, African dust, and wildland fires. The region, with a varied orographic climatology, shows high and diverse industrial activity. Urban and interurban road traffic of the region decreased by 49% and 53%, respectively, whereas industrial activity showed a lower reduction of 20%. Consequently, the average concentrations of NO_2_ in the cities during the period fell to 12.4 µg·m^−3^ (−45%). Ozone showed up to five exceedances of the WHOAQG for the daily maximum 8-h average in both rural and urban sites, associated with transport through France and the Bay of Biscay, under periods of European blocking anticyclones. However, averages showed a moderate decrease (−11%) in rural environments, in line with the precursor reductions, and disparate changes in the cities, which reproduced the weekend effect of their historical records. The PM_10_ decreased less than expected (−10% and −21%, in the urban and rural environments, respectively), probably caused by the modest decrease of industrial activity around urban sites and favorable meteorology for secondary aerosol formation, which could also influence the lower changes observed in the PM_2.5_ (−1% and +3% at the urban and rural sites, respectively). Consequently, in a future low NOx traffic emission scenario, the inter-regional PM and ozone control will require actions across various sectors, including the industry and common pollution control strategies.

## 1. Introduction

COVID-19, a disease caused by the SARS-CoV-2 coronavirus, expanded rapidly throughout the world during the first months of 2020, reaching pandemic status on March 11. Many countries were presented with unprecedented challenges with the intensive care units of their national health systems reaching saturation point. At this point (June 2021), the number of infections has exceeded 175 million and almost 4 million deaths have already been confirmed in the world (https://www.worldometers.info/coronavirus/; accessed on 14 June 2021). To reduce the spread of SARS-CoV-2, lockdown measures were implemented worldwide with varied timing and severity according to the onset of the epidemiological crisis and the evolution of infections. The reduction of mobility and changes in industrial/commercial activity affected global emissions, but many details of these changes are yet unclear. Annual national emission inventories previously lagged three years behind the present [1], and thus, the year 2020 is not yet available to assess the impacts of COVID-19. Present estimations of emissions can be based on mobility and industrial activity changes, electric and fossil fuel energy demand, economic output, and indirect evidence of air pollution changes (including ground-based and satellite observations) with respect to a previous reference period [2,3].

Using the year 2016 as the reference for the emission changes, Ref. [4] estimated average emission reductions for the period 20 January–26 April 2020 to be −33% for NO*_x_*, −8% for Non-Methane Volatile Organic Compounds (NMVOCs), −7% for SO*_x_*, and −7% for PM_2.5_ in the EU-28 plus Norway and Switzerland. These reductions vary according to the studied reference period and the analyzed countries, which were not subject to the same mobility–activity restrictions during the period. Activity and mobility in the EU-27 (EU-28 excluding the UK) were sharply reduced in the first half of March 2020 as a result of the implementation of lockdown measures which were initiated on February 21 in Lombardy (Italy) and then extended to the rest of the Union. Mobility gradually recovered in May in some areas faster than in others depending on how quickly lockdown measures were eased [5]. The reported shifts in human mobility and the industrial activity resulting from the lockdown offer a unique opportunity to identify their effects on urban and rural air quality.

As a result of the aforementioned emission reductions, many studies on air quality changes have been published. Some of them are focused on urban environments [6,7,8] or both urban and rural environments [9,10,11]. Such studies range from the local to the continental and global scale. Most of them estimate the average changes over a period for a selection of pollutants such as NO_2_, NO, O_3_, PM_10_, PM_2.5_, and CO_2_. In Spain, NO_2_ levels fell below 50% of the WHO annual air quality guidelines (WHOAQGs), but those of PM_2.5_ were reduced less than expected attributed to, among other causes, meteorological conditions favoring high secondary aerosol formation [11]. Decreases in PM_10_ levels were greater than in PM_2.5_. Daily maximum 8-h average (MDA8) O_3_ experienced a generalized decrease in all the rural receptor sites when lockdown restrictions were relaxed (June–July) with −20%. For urban areas (including Bilbao), the average MDA8 O_3_ period responses were heterogeneous, with increases or decreases depending on the period and location. However, the O_3_ WHOAQG was still exceeded during the lockdown in several cities. Sicard et al. (2020) also detected strong reductions in NO_2_ mean concentrations in four European cities in Italy, France, and Spain: ~53% at urban stations, comparable to Wuhan (57%), and ~65% at traffic stations. In Europe, NO declined even further, ~63% at urban stations and ~78% at traffic stations. Reductions in PM_2.5_ and PM_10_ at urban stations were much smaller. The NO_x_ concentrations during the lockdown were on average 49% lower than those at weekends of the previous years (2017–2019) in all cities. The lockdown effect on O_3_ production was ~10% higher than the weekend effect in the studied southern European cities. 

However, very few studies are devoted to pollution episodes and their analysis together with the observed emission/air pollution/meteorology variations inside or around the target domain, which could be important to understand the origin of the observed changes or to identify eventual transport episodes. In this respect, the evaluation of the concentration series at a daily/hourly resolution is needed, together with the observed meteorology/emission changes, in order to show the real length and intensity of episodes, search for their origin, and suggest actions for the eventual control strategies. In this manuscript, we show a comprehensive analysis of the impact of the COVID-19 lockdown on the air pollution of the Basque Country, a region in northern Spain with important industrial activity, and three major cities distributed throughout complex mountainous topography next to the French border. Special attention is paid to the occurrence of episodes during the period as well as their origin.

## 2. Materials and Methods

### 2.1. Area of Study and Selected Sites

The Basque Country (B.C.) is an autonomous Spanish community located in the north of the Iberian Peninsula, at the eastern end of the Cantabrian coast and near France (Figure 1). The orography of the B.C. is mountainous, with moderate altitude peaks of approximately 1000–1500 m.a.s.l (Figure 1). Three climatic zones can be distinguished: (1) the coastal zone and the North Atlantic slope, very rainy with moderate temperatures softened by the influence of the sea, where the main urban areas of Bilbao and Donostia-San Sebastián are located; (2) a transition zone between the oceanic and the Mediterranean climate, with mild temperatures and less precipitation than in the Atlantic slope, where Vitoria-Gasteiz is located; (3) the southernmost area corresponds to the Ebro valley, which drains into the Mediterranean, with important seasonal temperature fluctuations, dry and hot summers, and little precipitation throughout the year [12]. The climate of the coastal valleys and mountains can be classified as Cfb following the Köpen–Geiger climate classification, while the southernmost region shows a Csb-type climate [13]. Anticyclonic conditions have historically been reported to be behind the most severe pollution episodes in the region [14,15]. They are associated with weaker winds, calm, and/or re-circulations, which are also accompanied by a limited vertical dispersion of pollutants due to subsidence and the presence of temperature inversions at the lower troposphere. The subsidence inversions prevent pollution from efficient vertical ventilation and dispersion, as was already reported in the last century in the region [14,16,17] and can efficiently transport aged air masses with high ozone concentrations from more remote area sources into the studied region [18]. Anticyclones during the spring and summer seasons bring longer and more intense sunlight periods, which enhance ozone production after photochemical reactions from NO_x_ and NMVOCs precursors. For ozone in the Basque Country (B.C.), an efficient transport mechanism from the upwind regions of precursor emissions ideally occurs under high-pressure conditions in western Europe and southern France [18,19,20,21]. Easterlies and northeasterlies from the European landmass, when they persist for several days, are responsible for such transport. We documented that the local ozone concentrations can rise well above 120 µg·m^−3^ for the daily 8-h averages during consecutive days, giving rise to the most severe/long episodes registered in the regional monitoring network of the B.C. 

Almost half of the 2.2 million inhabitants of the B.C. live in the core and commuting zone of the city of Bilbao (the Greater Bilbao), the most populated city of the three provincial capitals. With 350,184 inhabitants in the core city, the rest are located in approximately 35 municipalities, most of them following the 15-km long estuary of the lower Nervión valley, from Bilbao city into the sea. The other two main capitals are Vitoria-Gasteiz with 253,996 inhabitants, and Donostia-San Sebastián with 188,240 inhabitants [22]. Regarding the economic activity of the whole region, industry represents 24.2% of the gross value in 2017 which was above the European EU-28 (19.6%) and the Spanish (16.2%) averages [23]. The most relevant sectors are metallurgy, machine tools, electricity generation, and transportation. The activity of the port of Bilbao represents 1.29% of the gross domestic product of the B.C. [24] and it is the most important Atlantic Port of Spain and the gateway to the European Atlantic trade routes. In Spain (312 veh·km^−2^), the province of Bizkaia ranks third for vehicle density, behind Madrid and Barcelona [11]. The province of Gipuzkoa also has a high vehicle density (238 veh·km^−2^), while Álava shows the lowest values (70 veh·km^−2^) of the B.C. [25]. 

The three provinces (Bizkaia, Gipuzkoa, and Álava), in Figure 1, show a different industrial distribution around their main cities (Figure 2). The location of the EPRTR industries [26] in and around the Greater Bilbao, follows the main estuary area and the two main valleys that drain into the city from the southeast and south. In Gipuzkoa, only a small fraction of its industry is located around the city of Donostia-San Sebastián, while the main fraction is well-distributed inside the three coastal valleys which drain directly into the sea, west of the city. In contrast, the main fraction of the industry in Álava is concentrated around the city of Vitoria-Gasteiz. The majority of the metal industry is located in the B.C., as shown in Figure 2, but there is also high diversity in industries distributed across the entire territory devoted to the production of chemicals and petrochemicals, cement, pulp and paper, waste treatment, waste incineration, and gas-fired power plants, among others.

The Air Quality Monitoring and Control Network managed by the Basque Government operates with the aim of evaluating the air quality in the whole territory. Currently, the network has 50 stations: 21 located next to industrial sources, 18 next to roads where traffic may be the main pollution source, and 11 at rural sites. Air pollutants are automatically measured on an hourly basis, as each analyzer is fitted with a suitable measurement technique and meets reference standards. This study is focused on regulated pollutants comprising sulfur dioxide (SO_2_), nitric oxide (NO), nitrogen dioxide (NO_2_), tropospheric ozone (O_3_), and particulate matter (PM_10_ and PM_2.5_) [27].

To assess the impact of the lockdown restrictions on the air pollution of the three provincial capitals, a selection of urban background sites and traffic stations was made, following the European Environmental Agency (EEA) criteria. This organization uses a viewer that tracks the weekly and monthly average concentrations of NO_2_ and PM https://www.eea.europa.eu/themes/air/air-quality-and-covid19 (accessed on 18 October 2021) to assess the effect of the COVID-19 lockdown upon the air pollution of European Countries. In this application, Bilbao (BI*) is represented by five stations and Vitoria-Gasteiz (VG*) and Donostia-San Sebastián (DS*) by four (Table 1). The Facultad de Farmacia station in Vitoria-Gasteiz was added to the initial three stations of the EEA list in order to include at least one ozone monitoring site in the city. Their respective locations are shown in Figure 1. “Cluster-station data” are used for each city, as in the EEA application, which represents averaged values for each urban site (BI*, VG*, and DS*), as shown in Table 1. Moreover, to complete the study, and to assess rural air pollution, three monitoring stations, one for each province, were also selected (Figure 1). Table 1 gathers the sites and the measured air pollutants on each station. 

In order to identify anomalous data, each individual hourly average concentration was tested and validated following an exhaustive procedure [28]. To avoid unreliable trends, daily, weekly, and yearly averages have been calculated only when more than 75% of the hourly data were available. For example, to obtain daily means, data with at least an 18 hourly average were needed

### 2.2. Selected Periods

In Spain, a state of alarm was established on Saturday 14 March 2020, and it was extended six times every two weeks until Sunday 21 June [29]. To study the lockdown impact on air quality in the B.C., 18 weeks were selected: 3 weeks before the lockdown (named w−3, w−2, and w−1), week 0 for the period 9–15 March, and the following 14 weeks of the lockdown (named w1–w14). Selected Monday-to-Sunday weeks, confinement, and mobility restrictions are all summarized in Table 2. 

Schools, colleges, and universities closed from 9 March onwards in Vitoria-Gasteiz, and from 13 March onwards in the remaining Basque territory. General restrictions enforced on 14 March included confinement and mobility limitations. Teleworking was enforced whenever possible, as non-essential services were ordered to close. In addition, from 30 March to 9 April, all non-essential service workers were forced to stay home. During this period, mobility was only allowed through force majeure, such as medical appointments or the acquisition of basic goods or food. 

Relaxation measures began on 4 May [29]. Mobility was restricted to the municipality and the use of public transport increased gradually. Allowance to leave was by age groups and time slots, and some establishments were open for appointments only, such as hairdressers, restaurants, bars, etc. Given the different situation of the Spanish Autonomous Communities, from 11 May onwards measures were dictated by the autonomous government [30]. In the following stages 1 and 2, social distancing measures were reduced progressively. From June 18 onwards, mobility between provinces was allowed and a situation of “new normality” started on 20 June [30]. 

In order to compare air quality data recorded during the 2020 lockdown (Table 2), corresponding periods for the five previous years were selected according to the criteria of the European Environment Agency [31], including the weekend effect. Selected Monday−to-Sunday weeks are detailed in Table 3. 

### 2.3. Emissions

The largest road traffic contribution in the B.C. corresponds to the province of Bizkaia (48–52% of the B.C. total in Table 4), and most of it corresponds to the Greater Bilbao’s urban area and its commuting zone. Data for the traffic changes during the lockdown (Section 3.1) were obtained from traffic counting stations at the three Basque Provinces [32] and as personal communication from each town council (Bilbao, Vitoria-Gasteiz, and Donostia-San Sebastián). Shipping emissions in ports are related to port traffic activity. The main port of the B.C. is in the estuary of the Greater Bilbao [33]. The contribution of the second port, in the province of Gipuzkoa, to the total traffic in the B.C. is marginal (7.7%), as shown in Table 4. The monthly variations on total traffic [34] have been used to estimate changes during the w0–w14 period (Section 3.1). The three airports in the B.C. are located 13–20 km from their respective core cities. The Vitoria-Gasteiz airport accounts for 99.8% of the B.C.’s merchandise (air goods), while the main airport of Bilbao accounts for 94% of passengers (Table 4). The estimation of aircraft emissions at ground level and their air pollution effect near airports have been analyzed in different studies. Indeed, the near surrounding areas of the airports, between 1 and 5 km, are the most sensitive to NO_x_ pollution, mainly emitted during the take-off and climbing phases. CO and Hydrocarbons (HC) are mostly emitted during the taxiing of aircraft, while SO_2_, PM_2.5_, and other trace compounds are primary air pollutants or precursors of secondary pollutants [35].

In the B.C. there are three natural gas combined cycle power plants with a total installed capacity of 2000 MW. They are all located in the commuting area of the Greater Bilbao: two of them at 14 and 17 km to the nothwest of the core city, and the third one to the southeast, following the valley of the Ibaizabal River, draining into the city (Figure 2a). Their emissions can impact the whole urban area under adequate meteorology. In the B.C., electricity from non-renewable sources comes from the three mentioned combined cycle thermal power plants (58%), a cogeneration plant (36%), and a waste incinerator (6%) [36], marked in closed red triangles in Figure 2a. 

Industrial activity and power plants are the main sources of NO_x_ in the region: they emit more than road traffic (5/4 ratio) and more than the residential domestic sources (5/1). They are also practically the only sources that emit SO_2_ and major emission sources of primary PM_2.5_ in a 4/1 ratio with respect to road traffic [37]. Annual emissions for the year 2018 (last update), and source distribution for NO_x_, SO_x_, PM, and NMVOCs, are shown in Figure 2. Industrial activity changes during the lockdown could be assessed by using the energy (electricity and gas) consumption changes discussed in Section 3.1. The Industrial Production Index (IPI) can also be used as an industrial activity indicator [38]

### 2.4. Meteorology

Meteorology is one of the main drivers of the pollutant concentration of a region. Intense winds bring ventilation periods with low concentrations while weak winds and/or local and mesoscale wind re-circulations give rise to the accumulation of pollutants and pollution episodes. A reduction of pollution concentrations in a region, like during the lockdown period, can be attributed to fewer emissions, good ventilation conditions, or both. Contrarily, an increase or the mere absence of a trend in a low emission scenario makes it necessary to search for adequate reasons, which could be related to non-accounted pollution sources and/or the concurrence of adverse meteorology. As it is discussed in Section 3.3, Section 3.4, and Section 3.5, this is the case for some of the pollutants monitored in the Basque network. To search for the role of meteorology in the observed concentrations during the lockdown period, meteorological anomalies with respect to the expected average conditions in the region are estimated in Section 3.2. Meteorological variables such as wind, cloud cover, temperature, and pressure distribution were selected to search for anomalies during the March–June 2020 lockdown period, with respect to the five-year 2015–2019 averages of the same four-month period. Hourly ERA-5 re-analysis data are used for the study [39], and the Grid Analysis and Display System (GrADS) for the data processing and representation of anomalies [40]. The six-hourly NCEP Climate Forecast System Reanalysis (CFSR) historical archive in Wetterzentrale (http://www.wetterzentrale.de/; last accessed: 14 June 2021) and the satellite imagery of NASA’s Global Imagery Browse Services (GIBS) were also used to analyze the main O_3_ and PM episodes recorded in the region during the lockdown in Section 3.5.

### 2.5. Air Quality Changes during the Lockdown

The concentrations of air pollutants recorded during the lockdown period in 2020 (Table 2) are compared with data obtained during the corresponding periods of the previous five years (2015–2019) in Table 3. Inter-annual and lockdown average concentrations of the full lockdown period, both at the selected three clustered urban sites and three rural sites, are estimated in Section 3.3 with their corresponding ensemble urban/rural averages. Hourly data availability is between 90% and 100% for all pollutants and sites, except for PM_2.5_, NO, and NO_2_ at the Pagoeta site during the 2015–2019 period, with 76%, 85%, and 85% of the data available, respectively, due to anomalous data for approximately three weeks which were removed from the database. The series of weekly concentrations during the lockdown can show changes with respect to climatological values not represented in the full lockdown averages. Comparisons at a higher time resolution (weekly or daily) of the NO_2_, O_3_, and PM_10_ concentrations during the lockdown with the corresponding 2015–2019 inter-annual values are performed in Section 3.4 and Section 3.5. They can help to study the effect of each stage of the lockdown in the air quality or to detect episodes of “anomalous” concentrations at this shorter time-scale above/below the inter-annual averages of the daily values. It is assumed that ozone and particles (natural or anthropogenic) episodes can occur randomly during the March–June period throughout the years. Those all events are included in the climatological baseline of daily averages and variability of the concentrations of the previous five years. In a scenario of low emissions, like during the lockdown, the positive anomalies can be related to transport episodes of ozone and particles (natural or anthropogenic) under adequate meteorology. The episodes last a variable number of days (usually less than a week) and the list of episodes [41] evaluated using the Navy Aerosol Analysis and Prediction System (NAAPS) with satellite imagery and high-resolution trajectories are used to search for positive anomalies associated with African dust, European sulfates, and wildfires over the climatological records (Section 3.4 and Section 3.5).

## 3. Results and Discussion

### 3.1. Mobility and Activity Reduction during the Lockdown

#### 3.1.1. Urban and Interurban Road Traffic

In order to study mobility decrease during the state of alarm in the B.C., the percentage of traffic reduction with respect to the same 2019 March–June period was calculated for urban and interurban roads (Table 4). Average urban and interurban traffic reduction in the B.C. during the lockdown was −49% and −53%, respectively. Its effect on air quality is discussed in Section 3.3. Traffic started to decrease during the first week with general restrictions (w1) and a minimum was reached during 6–12 April (w4), coinciding with restrictions on non-essential activities (Table 2). The decrease was significant for intercity roads (−82%) and down to −85% for highways for light vehicles with respect to w0 and less for heavy vehicle circulation on highways (−65%) as the transport of goods was an essential activity. From week 4 onwards, traffic tended to recover, but values previous to the lockdown were not reached. Regarding urban traffic, the largest reduction with respect to w0 was −81%, −70%, and −85% at Bilbao, Vitoria-Gasteiz, and Donostia-San Sebastián, respectively, during week 4. The time evolution of the traffic reduction is discussed in Section 3.3 in the context of the observed NO_2_ and O_3_ concentration changes in the three cities.

#### 3.1.2. Ports and Airports

The activity and logistics operations of the port of Bilbao, with the highest contribution, barely changed during the lockdown period (Table 4), ensuring the delivery of essential services and facilitating supply sources for industry and consumer products for customers [33]. Monthly variations on total traffic ranged between −3% (March) and −14% (June 2020) [34], with an average of −9.1% for the corresponding 2019 March–June period (Table 4). On the other hand, the three airports in the B.C. showed a decrease in their activity during the 2020 lockdown, although differently in each of them. Total passenger air traffic variation ranged between −91.0% and −94.4% at the three airports of the B.C., compared to the corresponding 2019 March–June period. The decrease in total merchandise air traffic affected Bilbao and Donostia-San Sebastián (−87.3% and −100%) more significantly. However, the Vitoria-Gasteiz airport, which accounts for 99.8% of the B.C.’s merchandise (air goods) traffic totals, barely changed during the lockdown period at −12.8% [42].

#### 3.1.3. Energy Demand: Electricity and Natural Gas

The observed changes in the electricity consumption by main sectors in the B.C. during the period March–June 2020 relative to the same period in 2019 are summarized in Table 5 (left column). The table is based on the monthly data of electricity consumption by sector (buildings, domestic, services, iron and steel industry, and rest of industry) forwarded by the Basque Energy Agency [43]. The electricity consumption decreased during the lockdown (−14.3%), dragged down by a decrease in the demand in industry and services, which was not balanced by the slight increase in demand (+4.8%) in the domestic sector after the “stay at home” order.

Between March and June 2020, with respect to the same months of 2019, the total consumption of natural gas decreased moderately (−11.1%), as shown in Table 5 (right column). This variation is supported by a moderate decrease in domestic and commercial consumption (−8.7%), a greater decrease in industrial consumption (−25%), and a significant growth for thermal power plants (+57%). 

#### 3.1.4. Industry

Industrial activity changes during the lockdown could be assessed by using the energy consumption changes by sector (Table 5): an average decrease between 18% and 22% during the period March–June 2020 with respect to the same period in 2019 could be inferred from the observed decrease in the industrial electricity demand. The Industrial Production Index (IPI) is also used as an industrial activity indicator. In the B.C., the IPI decreased by 21% during the period March–June 2020, with respect to the preceding months of January and February [38]. This decrease was similar (−20.4%) to that observed in Spain [44]. The estimated change was uneven among the different sectors of the industry, depending on the evolution of demand and the storage capacity of the industrial production, among other reasons. In this sense, the refinery located close to the port of Bilbao, the largest industrial emission source of the B.C. (Figure 2), maintained its full activity during the first two months of the lockdown and reduced it by 40% from the beginning of May onwards, due to the persistent low fuel demand after the drastic reduction in mobility [45]. On the contrary, the three natural gas combined cycle power plants increased their activity by 57% (Table 5), with the corresponding increase in their relatively important NO_x_ contribution.

Emission changes out of the B.C. during the lockdown need to be considered in our analysis because they can explain the transport of pollution from upwind regions. The increasing level of pollutant concentrations in the surrounding regions can contribute to local/regional episodes (see Section 3.5) under the appropriate meteorology. Following the IPI monthly activity indicator [44], the industrial activity in France, similar to that of Spain, decreased by 20.4% during the period March–June 2020 with respect to the preceding months of January and February. However, a lower decrease is found for the average industrial activity changes in the EU-27 member states (−16.4%).

Concerning urban and interurban road traffic, the largest mobility decrease in Europe, using mobile positioning data, was observed in Spain [5] with an average decrease of 55% during the period March–June 2020 relative to February before the lockdown, followed by France with an average decrease of 41%. The rest of Europe showed lower decreases. Ground transport CO_2_ emissions decreases were the largest in Spain (−16.6%), while France (−13.7%) and Italy (−13.0%) showed more modest decreases [46], although in this case the changes were estimated over the first seven months of 2020 with respect to the same period in 2019. 

### 3.2. Meteorology during the Lockdown Period (2020) and Comparison with 2015–2019

Meteorological anomalies during the lockdown period (16 March–21 June 2020) with respect to the five-year 2015–2019 averages of the same corresponding period are represented in Figure 3. ERA-5 re-analysis of temperature (shaded colors in °C) and cloud cover anomalies (contours in %) are presented in Figure 3a,b, while the mean sea level pressure (contours in hPa) and wind anomalies (shaded colors in m·s^−1^ and vectors) are included in Figure 3c,d. The lockdown period was characterized by positive temperature anomalies in the north and northwest of the Iberian Peninsula and negative temperatures over the south and southeast (Figure 3a,b). The highest temperature anomalies were located over the southwest of France and in “the most southeast region” of the Bay of Biscay. The same relative warm areas showed negative cloud cover anomalies, which also affected the B.C. region (Figure 3b). The pattern of pressure and wind anomalies observed in Figure 3c,d show more intense European continental easterlies blowing over the higher-pressure anomaly located in western France and offshore over the Bay of Biscay, while the wind intensity in the B.C. approaches the average values, with a slightly negative 0.5 m·s^−1^ anomaly. 

Consequently, in a low emission scenario, the actual ventilation conditions in the B.C. would have contributed to a comparative reduction of ambient concentrations of primary pollutants near the sources (Section 3.3). Pressure, wind direction, temperature, and cloud cover are compatible with a higher-than-normal frequency of anticyclones over France during the period: the continental E winds would have brought the observed warm and dry anomaly to the northern regions of the Iberian Peninsula. Sea breezes during the daytime, which are frequent from March to October in the Cantabrian Coast, are expected to contribute to the inland convergence of the European continental easterlies over the Bay of Biscay into the northern coast of Iberia [14,18,19]. All these processes are responsible for the transboundary transport of ozone (and secondary aerosols) from southern France, which adds to the local production to give rise to the most severe episodes registered in the region (Section 2.1). The changes in the lockdown period average values of PM and ozone between 2020 and previous years (Section 3.3) could be related to the mentioned meteorological anomalies (temperature, cloud cover, and winds). All the ozone episodes identified during the lockdown in Section 3.5 also show the synoptic forcing responsible for the observed anomalies in Figure 3.

### 3.3. Air Quality during the Lockdown, Full-Period Averages, and 2015–2019 Comparisons

#### 3.3.1. Inter-Annual Averages (2015–2019)

Due to the greater total traffic and industrial emissions in the Greater Bilbao (BI*) with respect to the other capitals (Section 2.3), its average inter-annual concentrations of pollutants (NO_x_, SO_2_, PM_10_, and PM_2.5_) were the highest among the three urban environments (Table 6). Similarly, DS* showed higher concentrations than VG*. This is true for most pollutants except for O_3_, which showed a reversed behavior, with lower concentrations at BI* due to an increased O_3_ titration by NOx. Inter-annual NO_2_ (27.2 μg·m^−3^) average concentrations in BI* (the highest among all the sites) were well below the annual WHO’s Air Quality Guideline (WHOAQG) for this pollutant (40 μg·m^−3^). The SO_2_ concentrations for the inter-annual values were also low (5.9 μg·m^−3^) and there was no exceedance of the 24-h average guideline (20 μg·m^−3^). However, PM_10_ (17.7 μg·m^−3^) and PM_2.5_ (9.5 μg·m^−3^) were close to exceeding their respective annual guidelines of 20 μg·m^−3^ and 10 μg·m^−3^. The largest urban O_3_ concentrations were registered in VG* (66.2 μg·m^−3^) and it was relatively low when compared to the rural averages. For the three rural sites MU, PA, and VA, the ensemble average of the inter-annual concentrations of NO (1.5 μg·m^−3^), NO_2_ (3.7 μg·m^−3^), and SO_2_ (1.9 μg·m^−3^) were much lower than those recorded in the urban sites (7.7, 22.6, and 3.9 μg·m^−3^ for the respective three pollutants). In this respect, the rural averages of NO and NO_2_ were only 19% and 16% of the respective urban concentrations while the rural SO_2_ were also relatively low: 49% of the urban values. This seems to be caused by the relatively large distance from their main sources (road traffic and industry) around the respective urban sites and the relatively short lifetimes of these pollutants. The average rural PM_10_ concentrations were 68% of the values registered in the cities. PM_2.5_ was even more uniformly distributed in the rural environment of the B.C. (5.6–6.2 μg·m^−3^), approximately 69% of the PM_2.5_ found in the average urban environment. The observed urban to rural concentration differences regarding NO and NO_2_ seem to be in accordance with the longer lifetime of PM_2.5_ and PM_10_, with resultant larger urban-to-rural differences at shorter lifetimes. On the contrary, the O_3_ field distribution was controlled both by the photochemistry and the availability of NO_x_ and NMVOC species. Thus, its concentration in the rural sites, out of the NO_x_- saturated urban environments, was higher than in the cities. The highest averages were found in VA (84.1 μg·m^−3^ in Table 6). This monitoring station showed the largest history of O_3_ episodes in the B.C. [21,47,48,49,50,51]. The occurrence of O_3_ exceedances of the WHOAQG (100 μg·m^−3^) for the MDA8 O_3_ concentration during the lockdown is discussed below, in Section 3.5.

#### 3.3.2. Lockdown Averages

##### NO and NO_2_

During the lockdown, both NO and NO_2_ urban concentrations showed the largest decreases among all the registered pollutants in the monitoring stations of the B.C. The ensemble urban average in the three sites during the period March–June 2015–2019 was 7.7 µg·m^−3^ and 22.6 µg·m^−3^ for NO and NO_2_, respectively. During the lockdown, they decreased an average of 4.1 µg·m^−3^ for NO and 10.2 µg·m^−3^ for NO_2_, an important reduction of −53% and −45%, respectively, when compared to the inter-annual values. These average values were not equally distributed among the cities, with VG* showing the largest NO_2_ reduction (−53%) and BI* the lowest (−42%). These reductions are in accordance with the reported road traffic decrease (−49% and −53% in urban and interurban road traffic, respectively) in the whole B.C. (Table 4), which had a direct impact on the urban NO_x_. Other contributors to the observed NO_x_ changes could have been the reported variations in the industrial and domestic commercial activity (Section 3.1), which use natural gas as the main source of heat or electric power. However, based only on the reported moderate reduction of the total gas consumption (−11% in Table 5) during the lockdown. It can be concluded that changes in road traffic were the main driver of the NO_x_ urban decrease.

The rural inter-annual concentrations of NO and NO_2_ in the B.C. were low because of their relatively short lifetime and long distance from the sources, as discussed above. Ensemble averages of the three sites were 1.5 µg·m^−3^ and 3.7 µg·m^−3^ for NO and NO_2_, respectively. During the lockdown, NO_2_ showed an important reduction of approximately 1.5 µg·m^−3^ in all three rural sites, while NO concentrations showed lower changes around its already very low value, with greater effect in MU than in VA and PA.

##### SO_2_


SO_2_ concentrations were very low in both urban and rural sites: the ensemble urban average during the period February–June 2015–2019 was 3.9 µg·m^−3^, and 1.9 µg·m^−3^ in the VA rural site. The urban SO_2_ was significantly higher in BI* (5.9 µg·m^−3^) and lower in VG* (2.7 μg·m^−3^) and DS* (3.2 µg·m^−3^). The observed distribution could be attributed to industry, which is the main source of ambient SO_2_ in the region (Section 2.1), with the relative higher impact of a large refinery located close to the mouth of the estuary of BI*, together with the ship traffic in its port area. During the lockdown, SO_2_ showed a reduction of −0.8 µg·m^−3^ in the urban sites and stayed constant in the rural ones. This represents a decrease of 21% in the cities for inter-annuals, which is in accordance with the estimated average 20% reduction in the activity of the refinery and the reduction (−18% to −25%) of the general industrial activity, based on their energy consumption. Traffic reduction in the port of Bilbao during the lockdown (−8.1% with respect to the same 2019 March–June period), could also have contributed to the observed SO_2_ decrease.

##### PM_10_

The PM_10_ lockdown average concentrations, as for the case of SO_2_, showed a moderate reduction in the urban and rural environments of the B.C. The ensemble average in the three urban sites during the period February–June 2015–2019 was 16.5 µg·m^−3^ and 11.2 µg·m^−3^ in the rural one. During the lockdown, PM_10_ decreased an average of −2.4 µg·m^−3^ and −1.7 µg·m^−3^ in the rural and urban environments, respectively. This represents a small reduction of 10% in the cities, which is lower than the observed one for SO_2_, and a larger reduction of 21% in the rural PM_10_ for inter-annual values. Reduced road traffic emissions, less construction/demolition activities, and the more modest industrial activity decrease (approximately −20%) could be the cause of the observed decrease. Thus, during the lockdown the decrease of the PM_10_ concentrations affected the rural environment more, resulting in an increased urban-to-rural concentration gradient. The reduction did not affect all cities or all rural sites equally: the lowest reduction of 3% in all the cities and 10% in all the rural sites corresponded to DS* and MU, respectively. Both monitoring sites are located at the seashore, and the sea salt aerosol could have played a role in the observed smaller reductions [11]. Sea salt aerosol contributed to a fraction of the total concentration, which probably did not change during the lockdown for inter-annual values, as can be inferred from the absence of wind anomalies in the coastal area of the B.C. (Figure 3). 

##### PM_2.5_

PM_2.5_ average concentrations during the lockdown stayed constant without significant changes for inter-annual values in both urban and rural environments of the B.C. For PM_2.5_, the ensemble inter-annual average in the three urban sites during the period March–June was 8.4 µg·m^−3^ and 5.8 µg·m^−3^ in the three rural sites. During the lockdown, similar concentrations were found both in the ensemble urban (8.3 µg·m^−3^) and rural (6.0 µg·m^−3^) averages. Moreover, concentrations remained similar at every specific site during the lockdown for their inter-annuals, except for the rural site MU which increased slightly (+13%) over a relatively low inter-annual value (6.2 µg·m^−3^). As shown in Section 3.2, wind velocity during the lockdown, which controls the ventilation/dispersion of pollutants, was close to the 2015–2019 average conditions. Thus, in a first approach, the lockdown values could be directly comparable with their inter-annuals without a wind velocity correction. 

After observing the quasi-persistent PM_2.5_ concentrations, with respect to inter-annuals, it was determined that urban and interurban road traffic could not be the main source for the observed PM_2.5_ concentrations in the B.C. because road traffic decreased significantly (49–53%). A significant PM_2.5_ reduction would have had a major impact on the cities, which was not the case. The observed reduction could only be attributed to a main PM_2.5_ source that did not change so much during the lockdown. In this respect, a spatial and sectorial source allocation study performed in 150 European cities, including the Greater Bilbao, using an adapted chemical transport model [37], showed that road transport contributes only 7% of the PM_2.5_ in Bilbao, while the industry contribution is 46%. The rest is distributed among agriculture (16%), residential (4%), natural (12%), and other minor sources. These percentages show high variability among the reported cities. For the case of Bilbao, which shows one of the largest industrial contributions of the European cities, more than 50% of the PM_2.5_ (primary and secondary) originated inside the Greater City, which could be helped by adopting the efficient local reduction policies on PM_2.5_ concentrations. During the lockdown, the main decrease affected road traffic, and industry showed a lower decrease, whereas, the activity of power plants increased (Section 3.1). Taking into consideration road traffic and industry only, as agriculture did not change significantly during the lockdown because the activity of agri-food companies was guaranteed, including crop and livestock holdings, we could expect a decrease of −1.2 µg·m^−3^ in Bilbao during the period, but the actual decrease was only −0.2 µg·m^−3^. The impact of specific meteorological conditions such as less rainfall and more insolation than during average March–June (2015–2019) meteorology (Figure 3), could have played a role in the more efficient formation of secondary aerosol, which could increase PM_2.5_ concentrations to the observed values [11]. The more uniform urban/rural PM_2.5_ distribution in the B.C. with respect to the PM_10_ can be attributed both to the location of their main primary and precursor sources (industry, road traffic, and domestic, among others) inside or close to the greater cities and their lifetime differences with PM_10_, which allowed a more efficient transport of PM_2.5_ to the more remote rural sites. 

##### Ozone

The ensemble O_3_ average concentration of the three urban environments of the B.C. during the lockdown did not show significant changes with respect to the inter-annual value, though important differences were appreciated among the three cities. Inter-annual O_3_ concentrations in the urban sites showed an average value (60.1 µg·m^−3^) below one of the rural sites (82.1 µg·m^−3^) with the lowest concentrations in BI* (51.5 µg·m^−3^), due to a greater effect of the O_3_ titration with respect to DS* (62.7 µg·m^−3^) and VG* (66.2 µg·m^−3^). During the lockdown, a significant increase was observed in BI* (+11%) and a moderate decrease in VG* (−7%) and DS* (−2%), which accounts for the observed quasi-unchanged average of the three sites (60.2 µg·m^−3^) concerning the inter-annual average (60.1 µg·m^−3^). On the contrary, a more uniform decrease (approximately −11%) was observed in the three rural environments making the differences between urban (60.2 µg·m^−3^) and rural (73.2 µg·m^−3^) sites smaller. The observed changes in the urban concentrations of O_3_ during the lockdown mimics the weekend effect in the three cities out of the lockdown period, as discussed in the next section.

Summarizing the results shown in Table 6, the average air pollutant concentrations during the March–June 2020 lockdown period with respect to the March–June 2015–2019 concentrations show different trends in urban and rural sites depending on both site and pollutant. At the urban sites, average concentrations of most air pollutants trend to decrease affecting NO and NO_2_ (−53% and −45%, respectively) more, SO_2_ and PM_10_ less (−21% and −10%, respectively), with little change for PM_2.5_ (−1%) and O_3_ (0%). The highest concentrations during the lockdown, as for the inter-annual averages, were registered in BI* and the lowest in VG*, except for O_3_, which showed a reversed behavior with a minimum at BI*. The observed decrease of pollution in the rural sites during the lockdown affected NO (−20%) and NO_2_ (−41%) resulting in very low concentrations of 1.5 µg·m^−3^ and 3.7 µg·m^−3^, respectively. The reduction was also important for SO_2_ and PM_10_ (−21%), moderate for O_3_ (−11%), and non-significant for PM_2.5_ (−1%).

### 3.4. Changes in the Air Quality of the Cities: Weekly Averages and Weekend Effects

The urban traffic decrease had a significant impact on the observed reduction of NO_x_ concentrations, as discussed in Section 3.3. PM_10_ and O_3_, among other pollutants, were also affected by traffic in the cities. NO_2_ weekly series for BI* (Figure 4a) and VG* (Figure 4b) are shown during both the lockdown period and the corresponding 2015–2019 period together with the percentage change in urban traffic during the lockdown in each city (dashed lines). The selection corresponds to the environments with the highest (BI) and the lowest (VG) pollution levels of the three cities. The NO_2_ full lockdown period average decrease in the cities (Table 6) is also well represented at the weekly scale for the full period. The lowest NO_2_ weekly average concentration in BI* corresponded to w7 with 11.1 µg·m^−3^, which was also a minimum in VG* (4.0 µg·m^−3^). However, these lowest concentrations are concurrent with the first week of the relaxation of the confinement and the mobility limitations, with the corresponding traffic recovery in the cities. The lower w7 values are more related to a period of favorable ventilation conditions with intense westerly winds, overcast conditions, and rain on the northern coast of Iberia. It is important to notice that, due to their respective rate of emissions, the lowest 11.1 µg·m^−3^ in BI* are above most of the weekly averages in VG* during the whole lockdown period (w2 to w14). The daily averages during the lockdown and the inter-annual 2015–2019 values with their standard deviations are also depicted for the same two cities BI* (Figure 4c) and VG* (Figure 4d). Inter-annual NO_2_ concentrations in VG* (dashed line) are well below the WHOAQG for the annual mean (40 µg·m^−3^) and during the lockdown they were even lower. However, the inter-annual concentrations in BI* are close to the WHOAQG; sometimes they exceeded it and, most of the time, the upper sigma interval is above the WHOAQG limit, at least until May-June, when its seasonal decrease is more evident. During the lockdown and after the road traffic decrease, the NO_2_ 24-h concentrations in BI* sank below the lowest values of the normal variability of the inter-annual values (gray shading in Figure 4c) and remained well below the WHOAQG values.

As in Figure 4a,b, for NO_2_, the weekly PM_10_ series for BI* and VG* are represented in Figure 5c,d. The generalized decreased concentrations observed in the three cities during the full lockdown period (Table 6) was also registered for all the weekly averages in Figure 4a,b, except for weeks w1–2, when the full lockdown and mobility restrictions were already active, and w11 during the relaxation stage. In these three weeks, simultaneous increases above the inter-annual values were registered in three cities. As discussed in Section 3.5, these periods were concurrent with African dust episodes. The O_3_ series for BI* and VG* are shown in Figure 5a,b. O_3_ concentrations in VG* during the lockdown (Figure 5b) showed a general trend to decrease with respect to the weekly inter-annual values in the same panel, except for weeks w2–3 and w10–11, which were concurrent with transport episodes of ozone from peripheric regions and under specific meteorological conditions (Section 3.5). In contrast, weekly ozone in BI* (Figure 5a) showed a trend to increase along the period of highest road traffic decrease in the city (w1-to-w12 in Figure 4a), with the exception of w7, with the above-mentioned exceptional meteorology, which resulted in a highly depleted O_3_ photochemistry. The same weekly series in DS* (not represented in the figure) showed no significant changes during the lockdown for inter-annual values. Thus, every city showed a different response in O_3_ concentrations to reduced traffic, despite being subject to the same restrictions and being located a short distance from each other (60–80 km). This may be related to the O_3_ chemical regime in each city. After interpreting their respective NO_x_ pollution levels, discussed above, and the O_3_ response to the traffic reduction, we interpreted the O_3_ increase in BI* as an indicator of O_3_ formation in NO_x_-saturated environments. In the same way, VG* is NOx-limited and DS* can be either limited or saturated depending on the specific situation. Unfortunately, the lack of NMVOC monitoring limited our capacity to fully interpret O_3_ changes at the urban locations. The analysis of the weekend effect in the three cities during the recent historical (2015–2019) series shows that O_3_ changes during the lockdown followed similar (positive–negative) trends than those observed between weekdays (Monday-to-Friday) and weekends (Saturday and Sunday) of the historical series. Figure 6 shows the hourly averages of O_3_ during the mean week (Monday to Sunday) of the lockdown period (solid red line) together with the corresponding inter-annual averages (dashed black line) and the standard deviations (gray shading) for the three clustered urban stations BI* (b), DS* (c), and VG* (d): the hourly O_3_ concentrations in BI* during the lockdown were, most of the time, above the represented range of the five-year inter-annual variability (gray shading); in DS* they were inside that range, while in VG* were below. In BI* these changes led to average increases of +7% and +5% in the MDA8 concentrations for the lockdown weekdays and weekends, respectively, compared to the inter-annual values (Figure 6a). Corresponding decreases of −8% and −10% were observed in VG*, while DS* showed the smallest changes (−3% and −1%). Similarly, the historical (2015–2019) weekend changes (Figure 6b) record an average increase in BI* (from 73.6 to 76.8 µg·m^−3^), a decrease in VG* (from 86.0 to 84.9 µg·m^−3^) and almost constant values in DS* (82.4–82.9 µg·m^−3^). 

### 3.5. Ozone and PM Episodes during the Lockdown

Weekly averages of O_3_ during the lockdown, discussed in the preceding section (Figure 5), show some weeks of ”anomalous” high O_3_ concentrations above the inter-annual averages in the more “rural” of the three cities (VG*), which showed a generalized trend to decrease after the road traffic decrease of the lockdown (weeks w2–w3 and w10–w11). “Anomalous” high PM_10_ concentrations were also recorded for the weeks w1–w2 and w11 in the three cities. On the contrary, the NO_2_ weekly averages were kept well under their inter-annual values in all locations during the whole lockdown as shown in Figure 4 for the most polluted city (BI*) and the least polluted (VG*) in the B.C. These anomalous periods for O_3_ and PM hide transport episodes from outer regions and with a different origin, as discussed next. A representation of the series at a (daily) higher resolution uncovers the real length and intensity of the detected episodes and it shows new episodes of short duration out of the signaled weeks.

A depiction of the daily time series of the concentrations of O_3_ and PM helps to better discriminate the episodes, which for the case of O_3_, lasted only an average of 3–4 days in the B.C. [19,20]. Thus, the recorded concentrations have been compared with the WHOAQGs. Figure 7 shows the MDA8 O_3_ concentrations during the lockdown and a short period of 4 weeks before (pre-COVID-19) at the urban sites. Five O_3_ episodes of an average duration of 3–4 days and with the daily 8-h standard above 100 µg·m^−3^ were recorded simultaneously in all urban sites (episodes numbered 1 to 5 in the figure). The simultaneous exceedance of the standards was selected for the identification of both O_3_ and PM transport episodes. The same O_3_ episodes were also recorded at all rural sites (Appendix A), which reached even higher concentrations and exceeded the long-term objective of the daily 8-h standard in the EU (120 µg·m^−3^) during episode number 5.

All episodes occurred with an upper-ridge extending from northwestern Africa to the Iberian Peninsula and a high-pressure surface, which expanded over the Bay of Biscay and the British Islands and France (Figure 7). This forced easterly winds in the marine boundary layer over the northern coast of Spain as well as sea-breeze convergence into the northern coast. As discussed in Section 2.1 and Section 3.2, the described synoptic scenario was behind most of the O_3_ episodes in the B.C. [19], and the main fraction of the O_3_ impacting the B.C. during these events could be attributed to transport across southern France. [21] concluded that local emission reduction policies would have a limited effect on the reduction of O_3_ levels in the B.C., as has been confirmed during the lockdown. Out of these “short” episodes of O_3_ transport during the lockdown and during a relatively long period (w4 to w8) its concentration in the rural/urban sites remained below the inter-annual averages and even below the lower sigma interval depicted in shaded gray color in Figure 7 and Appendix A. The city of BI, the most NOx-saturated environment of the three cities, and, to a lesser extent, DS, showed concentrations close to the inter-annual averages during the same period w4-w8, which mimics a rise in the O_3_ concentrations similar to the weekend effect in BI. Consequently, the ensemble urban average (60.2 µg·m^−3^ in the three cities) of the whole lockdown period did not show changes for the inter-annual values (60.1 µg·m^−3^), hiding the different behavior of each city. Similarly, the rural ensemble background average (73.2 µg·m m^−3^) showed a decrease of −8.9 µg·m m^−3^ (−11%) for inter-annuals (Table 6), also hiding episodes more severe than in the cities.

The observed changes during the lockdown reinforce these results: (1) the relatively high impact on the whole B.C. of the O_3_ importation from the continental Europe during episodes (rural and urban areas), and (2) the high local O_3_ anomaly in the city of Bilbao, which seems to be related with a NOx-saturated area. The city kept out of the NOx- sensitive conditions even after the reported emission reduction during the lockdown. In this respect, it is important to note that the observed behavior of the O_3_ averages during the low emission scenario of the lockdown in the three cities (Table 6) reproduced the observed weekend effect of the three cities (Section 3.4): BI* recorded higher O_3_ concentrations, DS* did not show significant changes, and VG* showed a decrease.

Figure 8 shows the 24-h average concentrations of PM_10_ at the three main urban sites. Horizontal red lines mark the WHOAQG values for the PM_10_ concentrations: 20 μg· m^−3^ for the annual mean and 50 μg·m^−3^ for the 24-h mean. Up to three PM_10_ events (numbered 1 to 3 in the figure) with the daily averages above 20 µg m^−3^ were recorded simultaneously in all urban sites during the whole period. Simultaneously high concentrations can be related to transport from distant sources. In this case, they all were concurrent with desert dust outbreaks from northern Africa after the development of large Rossby waves over the region, and/or after the evolution of isolated low-pressures moving to northwestern Africa, as shown in Figure 8. Out of these events occurring in weeks w1–w2, w8, and w12, which pushed the 24-h average concentrations above the inter-annual values, PM_10_ concentrations were slightly under those averages and near the lower boundary of the sigma intervals. As a result, the full period averages also remained under the inter-annual values (Table 6) in the urban environments of the B.C., with a larger decrease in VG* (−16%), moderate in BI* (−12%), and with almost no variation in DS* (−3%). In contrast to the O_3_ concentrations, which show greater average values in the rural sites in comparison to the urban ones (Table 6), PM_10_ concentrations in the rural environments show values well below those registered in the urban sites during the lockdown. There is a complete absence of concentrations above the reference 20 µg· m^−3^ level, except for the MU location during week w1 (Appendix A). The more important decrease in PM_10_ with a more local regional origin in the three rural sites in comparison to the urbans has prevented the African transport events 1, 2, and 3 from causing the PM concentrations to exceed the level of 20 µg·m^−3^. The two extreme PM_10_ peaks during the pre-COVID-19 period (weeks w−2 and w−3) in Figure 8 and Appendix A, corresponded to a wildfire in Tineo (Asturias), west of B.C., for the w−3 episode, and an African dust episode simultaneously with a wildfire for the w−2 episode.

The 24-h average concentrations of PM_2.5_ of the three urban sites are shown in Figure 9. The recorded concentrations during the lockdown were close to the inter-annual averages in the figure, and most of the time, they remained inside the sigma interval depicted in shaded gray color. The horizontal red lines mark the WHOAQG values for PM_2.5_: 10 μg·m^−3^ for the annual mean, and 25 μg·m^−3^ for the 24-h mean. Similarly, for the identification of the PM_10_ episode, and using the annual WHOAQG for PM_2.5_ (10 µg·m^−3^), the same events registered in weeks w1–w2, w8, and w12 for PM_10_ emerge for the fine fraction (marked with an arrow in the figure), with the daily average above the WHOAQG simultaneously in all urban sites. In addition, more simultaneous peak events can be identified for weeks w3, w4, w6, w10, and w11 (not arrowed in Figure 9), which are not present in the PM_10_ record because they did not reach enough concentration in the three cities. Following the list of episodes in MITECO (2021), European sulfates were behind the first three PM_2.5_ events (w3, w4, and w6), and w10 was an African dust episode. None of them were identified in the PM_10_ records. The recorded African dust episodes during weeks w1–w2 and w12 were also identified by the system together with concurrent PM from wildfires. All the episodes identified by the surveillance system for the B.C. during the period are clearly represented in the PM_2.5_ records and vice-versa with the exception of the peak in week w11, which was recorded simultaneously in the three cities but not identified by the system. This unexplained peak is concurrent with an O_3_ transport episode discussed above, and it could be related with (non-sulfate) secondary aerosols generated by photochemical reactions, and transported together with O_3_ into the B.C. Similar high 24-h averages, close to inter-annual values, were observed for PM_2.5_ in the rural sites (Appendix A). The observed relatively high daily PM_2.5_ concentrations are in line with the calculated period average concentrations in Table 6, which show small changes with respect to the inter-annual values, both in rural and urban environments. Contrary to PM_10_, which showed a more drastic reduction in concentration in the rural sites during the lockdown, the more uniform distribution of PM_2.5_ (Table 6) contributed to reaching the reference concentration (10 µg·m^−3^ ) in the three rural (Appendix A) and urban sites (Figure 9) during weeks w1-w2 and w11. The rest of the recorded PM_2.5_ peaks laid, approximately, at the reference 10 μg·m^−3^ level, but without exceeding it simultaneously in the three rural sites. The two extreme PM_2.5_ peaks during the pre-COVID-19 period (week w−2) in Figure 9 and Appendix A correspond to the same wildfire and African dust episodes also observed in the PM_10_ records (Figure 8 and Appendix A).

## 4. Summary and Conclusions

We have evaluated the impact of the lockdown during the COVID-19 pandemic for the three main urban sites of the B.C. region and their adjacent rural sites during the period of confinement and mobility restriction (14 weeks). We selected 16 stations of the Air Quality Monitoring and Control Network which represented the whole territory. As described in many other regions of the world including the rest of Spain [11], the concentrations of most pollutants decreased in the B.C. due to COVID-19 restrictions, with the urban and interurban road traffic of the region falling an average of 49% and 53%, respectively, and a lower reduction of approximately 20% of the general industrial activity. The observed changes of the full-period averages, however, were not uniform throughout the territory, and they varied with each pollutant. In addition, the whole territory was repeatedly affected by aerosol and O_3_ episodes transported from outer regions and added to the local pollution. They occurred under specific meteorological conditions.

At the urban sites, the observed decrease of pollution during the lockdown affected NO and NO_2_ more than any of the other registered pollutants (−53% and −45%, respectively) resulting in a very low concentration for the ensemble urban NO_2_ average (12.4 µg·m^−3^) well below the WHOAQG for the annual average of 40 µg·m^−3^. In the rural sites, a similar reduction affecting the already low inter-annual values of these sites resulted in very low concentrations of both NO (1.2 µg·m^−3^) and NO_2_ (2.2 µg·m^−3^). The observed decline was more in accordance with the reported road traffic decrease, which has a direct impact on the urban NO_x_, and less with the more limited reduction of the industrial activity as discussed above. The decrease was even more moderate for SO_2_ (−21%) in both rural and urban environments, in line with the moderate reduction of the general industry, although this decrease affected their already very low inter-annual concentrations (3.1 and 1.5 µg·m^−3^ for the urban and rural environments, respectively). 

During the lockdown, PM_10_ decreased moderately (−10% and −21% in the urban and rural environments, respectively). Inter-annual period averages in BI* (15.5 µg·m^−3^) are close to the WHOAQG for the annual averages (20 µg·m^−3^). Alternatively, during the lockdown, the most uniform concentration distribution was registered for PM_2.5_ (8.3 µg·m^−3^ and 6.0 µg·m^−3^ in urban and rural sites) and it also showed the lowest changes with regards to pre-pandemic values (−1% and +3% at the urban and rural sites, respectively). The greater average concentrations in BI* (9.5–9.3 µg·m^−3^) places this location very close to the WHOAQG for the annual averages (10 µg·m^−3^), even during the lockdown. Apart from the relative weight of the local industry emissions, which could have contributed to the observed modest changes in the PM_2.5_ concentrations, the impact of the documented meteorological anomalies during the period (less cloud cover than during the average March–June 2015–2019 period) could have also contributed to the more efficient formation of secondary aerosol, and the subsequently relatively high PM concentrations. 

O_3_ concentrations showed low changes in the inter-annual averages in the cities (ensemble average 0%) but, unlike PM_2.5_, showed important differences among the three cities: BI* increased (+11%), DS* remained almost unaltered (−2%), and VG* decreased (−7%). These changes occurred in the same direction as the weekend effect already observed in historical data. The changes in the rural sites showed a moderate decrease (−11%) from the relatively high average concentrations (82.1 µg·m^−3^) of the inter-annual records. This is in line with lower emissions of anthropogenic precursors. The widespread decrease in the rural sites did not avoid the reported five O_3_ episodes during the period and the exceedance of the long-term objective of the daily 8-h standard (120 µg·m^−3^) in all these sites. Simultaneous exceedances of the WHOAQG for the MDA8 of the O_3_ concentrations were also observed in the three cities. These episodes are attributed to transport from France of important background levels of ozone under anticyclonic conditions, which added to the local production. 

We did not find exceedances of the WHOAQG for either PM_10_ and PM_2.5_ 24-h average concentrations during the lockdown. However, frequent episodes of consecutive PM_2.5_ daily values above the annual WHOAQG were identified and less frequent for the PM_10_. African dust transport, wildland fires, and anthropogenic European sulfates, together with more frequent spells of favorable meteorology for secondary aerosol formation at the local regional scale, seem to have been behind the observed exceedances.

The observed air quality response to the activity and road traffic emission reductions during the lockdown period suggests that the results of having a high proportion (above 50%) of the vehicle fleet switching to electric in the near future will affect the air quality of the B.C., based on the NO_x_ emission reductions. However, the reduction of fine and ultrafine fractions of particles, due to the use of electric vehicles, important from the point of view of their number, not their mass, can mean a non-negligible improvement from a health point of view [52,53]. Sustained improvements in the PM_10_, PM_2.5_, and O_3_ of the region will require actions across various sectors, including industry, together with inter-regional European initiatives to establish pollution control strategies based on inter-regional policies. 

The limited number of monitoring stations used for the current study does not represent the whole region in full, though the selected clustered urban sites characterize the main three core cities and their peripheral commuting zone, where the main fraction of the B.C. population resides. Thus, the caveat of our assessments on the impact of the COVID-19 lockdown upon the air quality is that outside the reach of the major cities there may be smaller villages where the relatively low decline in the local industrial activity could have resulted in non-significant changes in the local air pollution. This could be important when considering diffuse emissions, odor impacts, or pollutants not registered in the monitoring network.

## Figures and Tables

**Figure 1 ijerph-18-11042-f001:**
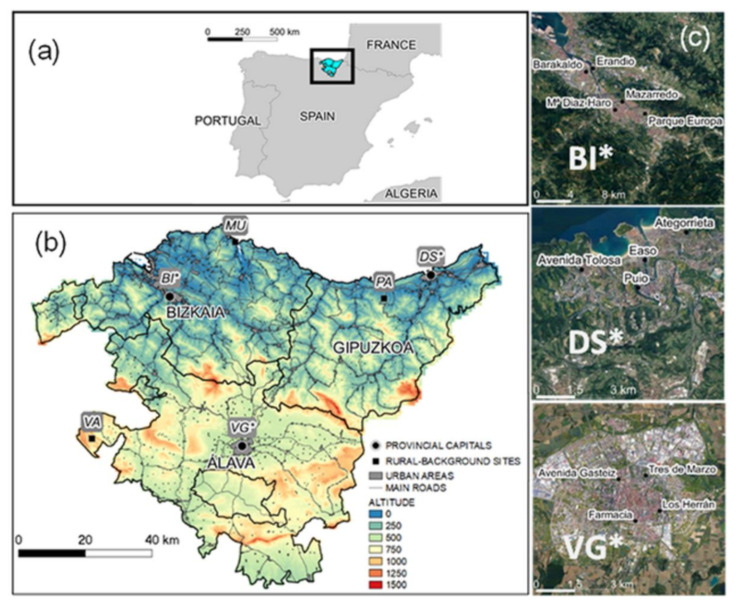
(**a**) Location of the Basque Country in Spain; (**b**) Topographic map. Colored scale indicates altitude (m) above the mean sea level (a.m.s.l.), bold lines separate the provinces (Bizkaia, Gipuzkoa, and Alava), main urban areas are shaded in gray and main roads are marked with thin solid lines. Measurement sites correspond to provincial capitals, clustered urban sites: Bilbao (BI*), Donostia-San Sebastian (DS*), and Vitoria-Gasteiz (VG*), and rural sites: Mundaka (MU), Pagoeta (PA), and Valderejo (VA); (**c**) Monitoring sites for each clustered urban traffic station (BI*, DS*, VG*).

**Figure 2 ijerph-18-11042-f002:**
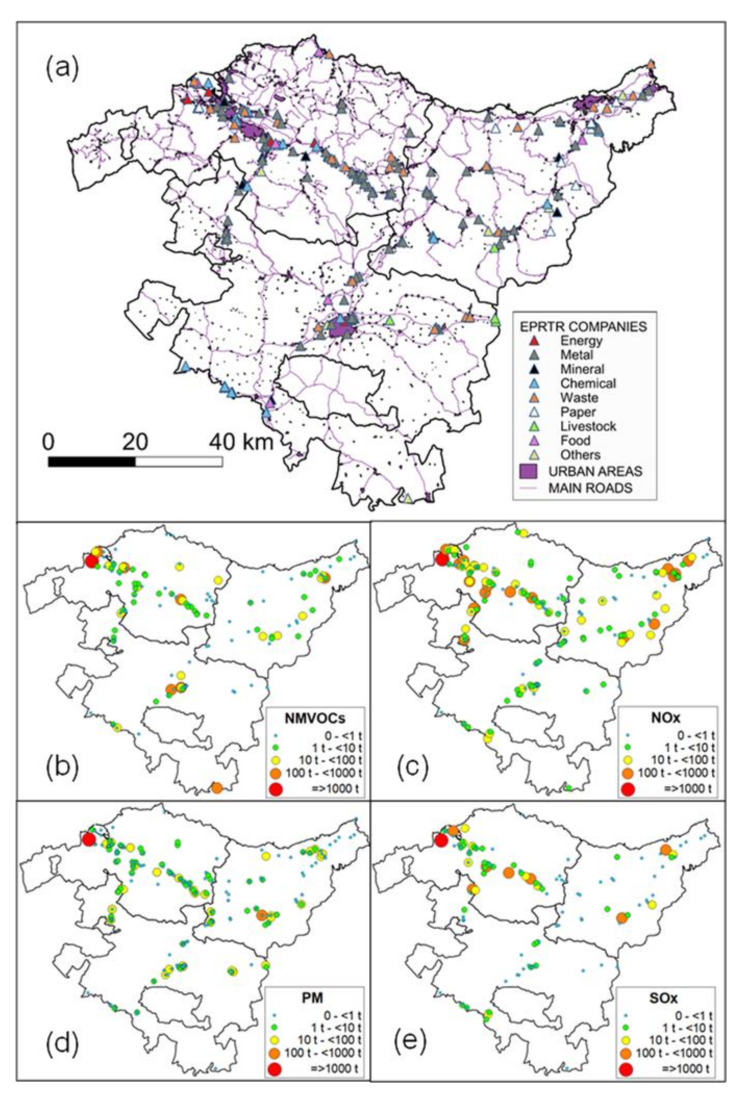
(**a**) Location of the companies registered in the European Pollutant Release and Transfer Register (EPRTR), Basque Country (2018), main urban areas, and roads. (**b**–**e**) Estimated annual emissions of NMVOC, NOx, PM, and SOx, respectively, from companies registered in the EPRTR.

**Figure 3 ijerph-18-11042-f003:**
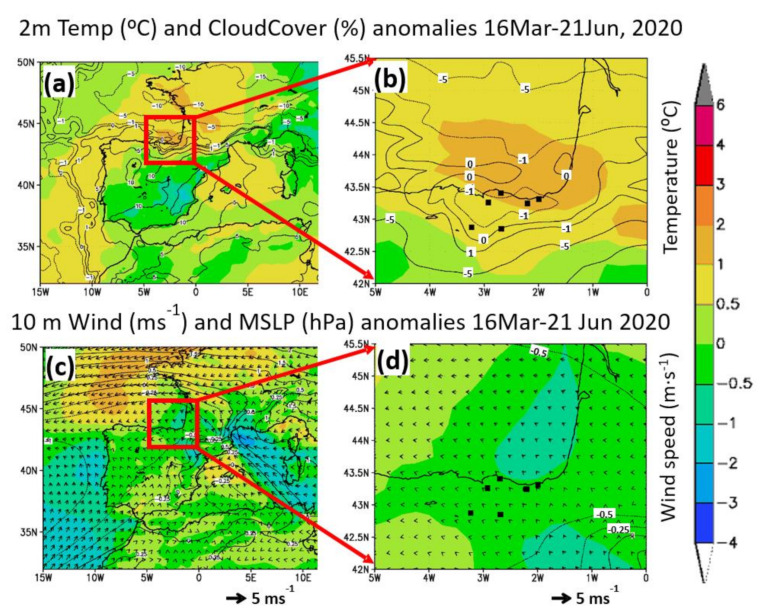
Temperature, cloud cover (**a**,**b**), mean sea level pressure, and wind anomalies (**c**,**d**) during the 2020 lockdown period with respect to the 2015–2019 corresponding period.

**Figure 4 ijerph-18-11042-f004:**
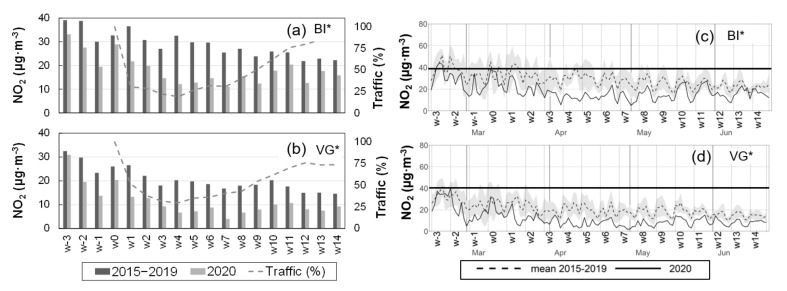
(**Left**) Weekly NO_2_ averages during the March–June (2020) lockdown period and the corresponding 2015−2019 inter-annual averages for the urban BI* (**a**) and VG* (**b**) monitoring stations. Dashed lines in the same panels represent the percentage change in weekly traffic during the lockdown compared to week 0. (**Right**) NO_2_ daily averages during the lockdown (solid line) and the inter-annual averages (dashed) with the standard deviation (gray shading) for the same two cities BI* (**c**) and VG* (**d**).

**Figure 5 ijerph-18-11042-f005:**
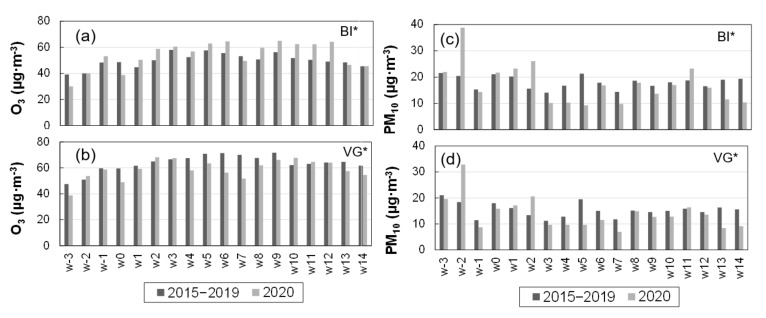
(**Left**) Weekly O_3_ averages during the March–June (2020) lockdown period and the corresponding 2015–2019 inter-annual averages in the urban BI* (**a**) and VG* (**b**) monitoring stations. (**Right**) Same type of representation for the PM_10_ weekly averages in BI* (**c**) and VG* (**d**).

**Figure 6 ijerph-18-11042-f006:**
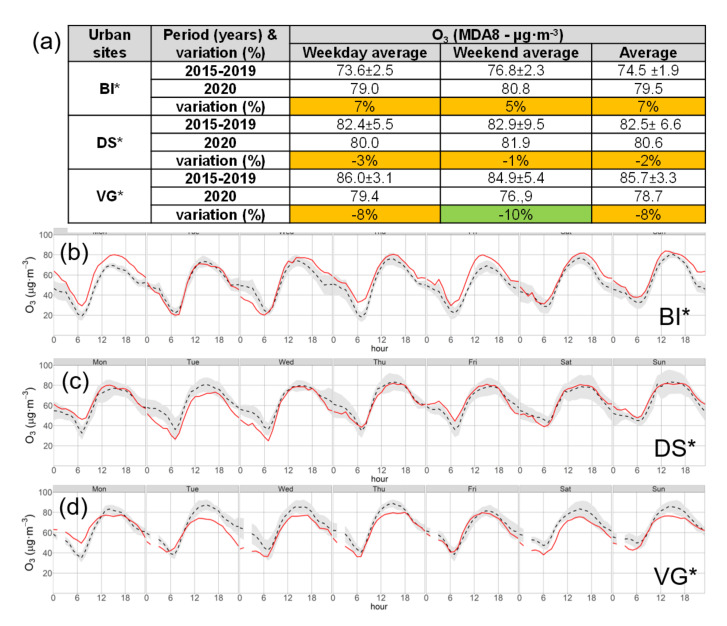
Daily maximum 8−h average (MDA) O_3_ values for weekdays, weekends, and the full period are shown in the table for the lockdown period and the corresponding years 2015–2019 in the three cities (**a**). The hourly averages of ozone during the mean week (Monday to Sunday) of the lockdown period are represented (solid red line) together with the corresponding inter-annual averages (dashed black line) and the standard deviations (gray shading) for BI* (**b**), DS* (**c**), and VG* (**d**).

**Figure 7 ijerph-18-11042-f007:**
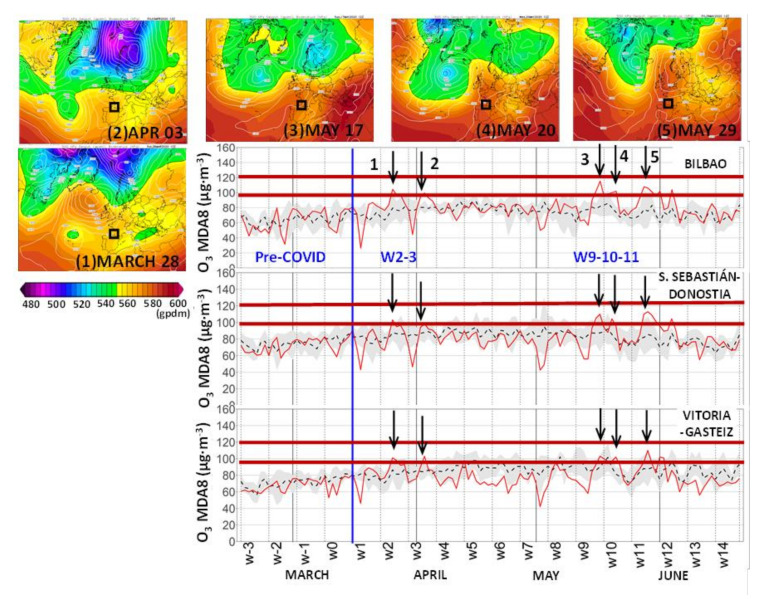
MDA8 ozone daily series during a pre−COVID19 four-week period and the lockdown period (solid red line) in the three urban sites. The series is represented in Figure 4c,d for NO_2_. Inter-annual averages (dashed) with their standard deviation (gray shading) can be compared with the lockdown daily values (solid red). Five simultaneous O_3_ exceedances of the WHOAQG of 100 μg·m^−3^ (horizontal red line) are marked (arrows) during weeks 2, 3, 9, 10, and 11 of the lockdown period. The synoptic forcings of the respective episodes are shown (upper panels) in the Climate Forecast System Re-analysis (CFSR), one panel per episode: the 500 hPa geopotential heights (gpdams) and mean sea level pressure (MSLP) contours (hPa) at 12:00 UTC are depicted (Source: Wetterzentrale).

**Figure 8 ijerph-18-11042-f008:**
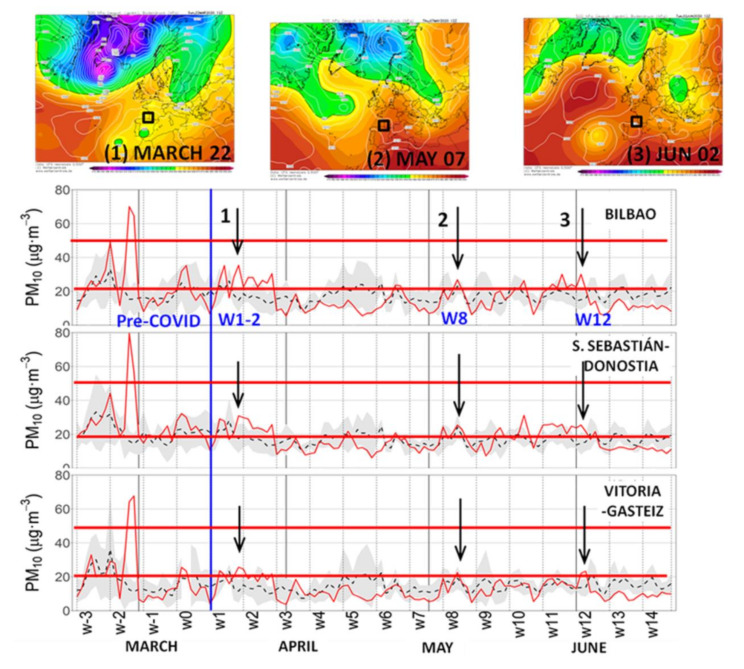
24-h average concentration series of PM_10_ at the three main urban sites. As in Figure 7**,** inter-annual values (dashed lines) with their standard deviation (gray shading), and the lockdown daily values (solid red) are represented. Three PM_10_ episodes, above the reference level of the WHOAQG of 20 μg·m^−3^, for the annual average concentration (horizontal red line), are marked (arrows) during weeks 1–2, 8, and 12 of the lockdown period. The respective synoptic forcings are shown (upper panels) in the Climate Forecast System Reanalysis (CFSR), one panel per episode, as in Figure 7.

**Figure 9 ijerph-18-11042-f009:**
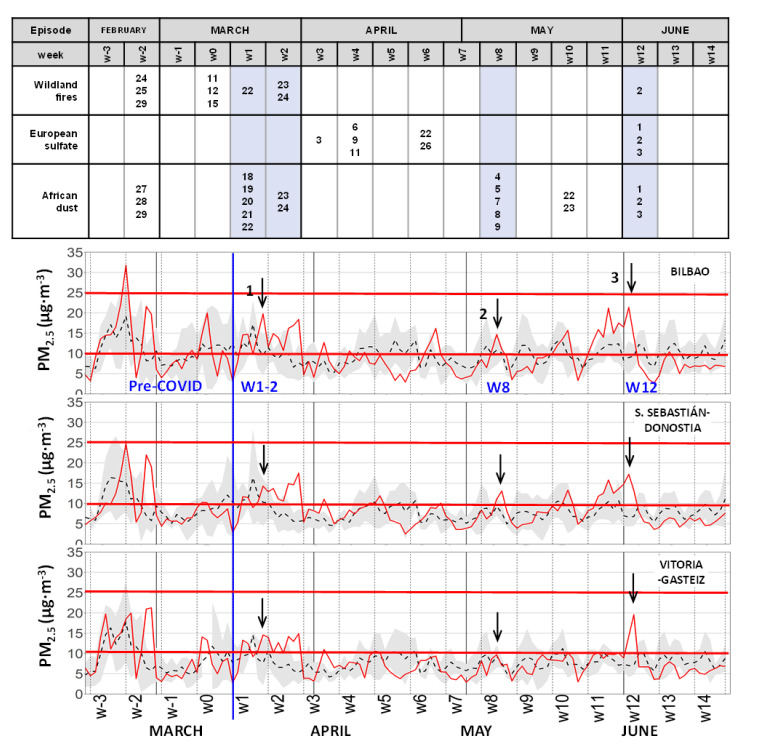
24−h average concentration series of PM_2.5_ at the three main urban sites. As in Figure 8, inter-annual values (dashed lines) with their standard deviation (gray shading), and the lockdown daily values (solid red) are represented. The same three PM_10_ episodes marked in Figure 8 (arrows) can also be detected in the PM_2.5_ series (above the WHOAQG of 10 μg·m^−3^). More episodes above the reference line (10 μg·m^−3^) can also be observed. The origin, date, and the pandemic week reference of each episode are shown in the upper panel table, as informed by the surveillance program of the Spanish Government, based in the Navy Aerosol Analysis and Prediction System (NAAPS) by the U.S. Naval Research Laboratory (NRL).

**Table 1 ijerph-18-11042-t001:** Selected urban background, traffic and rural sites, corresponding stations, and measured pollutants. Clustered urban background and traffic sites are marked with an asterisk *.

Sites	Station	Measured Pollutants
**Urban background and traffic sites**		
BI*	Parque Europa—Bilbao	NO, NO_2_, SO_2_, PM_10_, PM_2.5_, O_3_
	María Diaz de Haro—Bilbao	NO, NO_2_, SO_2_, PM_10_, O_3_
	Mazarredo—Bilbao	NO, NO_2_, SO_2_, PM_10_
	Barakaldo—Bilbao	NO, NO_2_, SO_2_, PM_10_
	Erandio—Bilbao	NO, NO_2,_SO_2,_ PM_10_, PM_2.5_
VG*	Tres de marzo—Vitoria-Gasteiz	NO, NO_2_, SO_2_, PM_10_, PM_2.5_
	Avenida Gasteiz—Vitoria-Gasteiz	NO, NO_2_, PM_10_, PM_2.5_
	Los Herrán—Vitoria-Gasteiz	NO, NO_2_, PM_10_, PM_2.5_
	Facultad de Farmacia—Vitoria Gasteiz	O_3_
DS*	Avenida Tolosa—Donostia-San Sebastián	NO, NO_2_, SO_2_, PM_10_, PM_2.5_, O_3_
	Puio—Donostia—San Sebastián	NO, NO_2_, SO_2_, PM_10_, O_3_
	Easo—Donostia-San Sebastián	NO, NO_2_, SO_2_, PM_10_,
	Ategorrieta—Donostia-San Sebastián	NO, NO_2_, PM_10_, PM_2.5_
**Rural sites**		
MU	Mundaka	NO, NO_2,_PM_10_, PM_2.5_, O_3_
VA	Valderejo	NO, NO_2_, SO_2_, PM_10_, PM_2.5_, O_3_
PA	Pagoeta	NO, NO_2,_PM_10_, PM_2.5_, O_3_

**Table 2 ijerph-18-11042-t002:** Weekly periods studied and mobility and confinement restrictions applied in the Basque Country.

Week	Weekly Period	Restrictions
w−3	17–23 February	None.
w−2	24 February–1 March	
w−1	2–8 March	
w0	9–15 March	Schools and universities close down. Confinement and mobility limitations.
w1	16–22 March	Confinement and mobility limitations.
w2	23–29 March	
w3	30 March–5 April	Confinement, mobility limitations, and restrictions to non-essential activities. Full lockdown: all non-essential activities stopped.
w4	6–12 April	
w5	13–19 April	Confinement and mobility limitations. Partial lockdown: some non-essential activities start.
w6	20–26 April	
w7	27 April–3 May	Relaxation of confinement and mobility limitations. Children under 14 years allowed to go out for 1 h a day.
w8	4–10 May	Relaxation and preparation—stage 0. All population is allowed to go out for 1 h a day in time bands according to their age, mobility is reduced to the municipality, and some non-essential services opened with capacity restrictions.
w9	11–17 May	Relaxation—stage 1. All population is allowed to go out, but mobility is reduced to the municipality.
w10	18–24 May	
w11	25–31 May	Relaxation—stage 2. Mobility is extended to the province.
w12	1–7 June	
w13	8–14 June	Relaxation—stage 3. Borders between provinces and other autonomous communities bordering B.C. were opened.
w14	15–21 June	

**Table 3 ijerph-18-11042-t003:** Corresponding selected pre-lockdown and lockdown weeks for the years 2015–2019 and 2020.

Year	Pre-Lockdown Weeks (w−3–w0)	Lockdown Weeks (w1–w14)
2015	16 February–15 March	16 March–21 June
2016	15 February–13 March	14 March–19 June
2017	13 February–12 March	13 March–18 June
2018	19 February–18 March	19 March–24 June
2019	18 February–17 March	18 March–23 June
2020	17 February–15 March	16 March–21 June

**Table 4 ijerph-18-11042-t004:** Contribution (%) and variation (%) of mobility in 2020 compared to 2019 in the three provinces of the Basque Country during the period studied (w0-w14) in different areas: interurban and urban traffic, air goods, passengers, and port traffic. **Large reduction (from −100.0% to −25.0%), reduction (from −24.9% to −10.0%), small reduction (from −9.9% to −0%)**.

Traffic	Contribution/Variation (%)	Bizkaia	Alava	Gipuzkoa	Weighted Average, BC
Road interurban	Contribution (%)	52	13	35	100
	Reduction (%)	−53	−58	−51	−53
Road urban (main city)	Contribution (%)	47	34	19	100
	Reduction (%)	−44	−49	−63	−49
Air goods	Contribution (%)	1	99	−	100
	Reduction (%)	−87	−13	−	−14
Air passengers	Contribution (%)	93	2	5	100
	Reduction (%)	−91	−94	−92	− 91
Port traffic	Contribution (%)	92	−	8	100
	Reduction (%)	−8	−	−21	−9

**Table 5 ijerph-18-11042-t005:** Variation of the energy consumption (electricity and natural gas) by main sector in 2020 relative to 2019, for the March–June average. **Large reduction (up to −25.0%), reduction (from −24.9% to −10.0%), small reduction or increment (from −9.9% to 10.0%), increment (above 10.1%)**.

Electricity Sector	Variation (%)	Natural Gas Sector	Variation (%)
Total electricity	−14	Total natural gas	−11
Industry (excluding iron and steel industry)	−18	Industry (excluding power plants)	−25
Iron and steel industry	−22	Power Plants	57
Buildings	−8	Domestic and Commercial	−9
Domestic	5		
Services	−18		

**Table 6 ijerph-18-11042-t006:** Average and ± standard deviation of NO, NO_2_, SO_2_, PM_10_, PM_2.5_, and O_3_ (μg·m^−3^) concentrations during the lockdown period (2020) and previous years (2015–2019) at clustered urban traffic sites (marked with an asterisk) and selected rural sites in Basque Country. Ensemble urban and rural averages are also included. The percentage of reduction or increment during the 2020 lockdown with respect to 2015–2019 is included in shaded colors: **Large reduction (up to −25.0%), reduction (from −24.9% to −10.0%), Small reduction or increment (from −9.9% to 10.0%), increment (above 10.1%). Clustered urban background and traffic sites are marked with an asterisk***.

Area	Station	Period (Years)/Variation (%)	NO	NO_2_	SO_2_	PM_10_	PM_2.5_	O_3_
Urban traffic	Bilbao (BI*)	2015−2019	8.4 ± 13.0	27.2 ± 15.7	5.9 ± 3.8	17.7 ± 9.6	9.5 ± 9.6	51.5 ± 26.6
		2020	5.0 ± 6.5	15.8 ± 9.8	4.9 ± 2.6	15.5 ± 8.9	9.3 ± 8.9	57.2 ± 25.9
		variation (%)	−40%	−42%	−17%	−12%	−2%	11%
	Donostia-San Sebastián (DS*)	2015−2019	9.7 ± 14.6	21.7 ± 13.9	3.2 ± 0.6	17.2 ± 8.3	7.9 ± 8.3	62.7 ± 24.9
		2020	3.9 ± 7.3	12.7 ± 8.4	2.6 ± 0.9	16.6 ± 8.1	8.1 ± 8.1	61.7 ± 24.2
		variation (%)	−60%	−41%	−19%	−3%	3%	−2%
	Vitoria-Gasteiz (VG*)	2015−2019	5.0 ± 8.8	18.8 ± 12.3	2.7 ± 0.7	14.7 ± 8.6	7.8 ± 8.6	66.2 ± 23.5
		2020	1.9 ± 3.4	8.8 ± 7.0	1.9 ± 0.7	12.4 ± 6.7	7.6 ± 6.7	61.6 ± 21.5
		variation (%)	−62%	−53%	−30%	−16%	−3%	−7%
	Ensemble urban average	2015−2019	7.7	22.6	3.9	16.5	8.4	60.1
		2020	3.6	12.4	3.1	14.8	8.3	60.2
		variation (%)	−53%	−45%	−21%	−10%	−1%	0%
Rural background	Mundaka (MU)	2015−2019	1.3 ± 0.8	4.2 ± 3.0	-	11.0 ± 8.0	6.2 ± 8.0	79.2 ± 18.6
		2020	0.9 ± 0.7	2.9 ±2.3	-	9.9 ± 6.6	7.0 ± 6.6	70.7 ± 19.1
		variation (%)	−31%	−31%	-	−10%	13%	−11%
	Pagoeta (PA)	2015−2019	2.0 ± 1.1	3.5 ± 2.6	-	12.7 ± 6.5	5.7 ± 6.5	82.9 ± 15.9
		2020	1.5 ± 0.7	1.6 ± 1.6	-	8.1 ± 6.0	5.5 ± 6.0	75.8 ± 17.3
		variation (%)	−25%	−54%	-	−36%	−4%	−9%
	Valderejo (VA)	2015−2019	1.1 ± 0.7	3.5 ± 1.9	1.9 ± 1.1	9.9 ± 6.5	5.6 ± 6.5	84.1 ± 21.3
		2020	1.1 ± 0.4	2.0 ± 1.4	1.5 ± 0.7	8.3 ± 5.0	5.6 ± 5.0	73.0 ± 21.5
		variation (%)	0%	−43%	−21%	−16%	0%	−13%
	Ensemble rural average	2015−2019	1.5	3.7	1.9	11.2	5.8	82.1
		2020	1.2	2.2	1.5	8.8	6.0	73.2
		variation (%)	−20%	−41%	−21%	−21%	3%	−11%

## Data Availability

All the public data and software used are contained in sources cited in the body text and in the references.

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
