# Peer review of "Impact of the COVID-19 Lockdown in a European Regional Monitoring Network (Spain): Are We Free from Pollution Episodes?"

_ijerph, 2021, doi:10.3390/ijerph182111042_

Round 1
Reviewer 1 Report
This is a well-organized and well-written draft.
PM2.5 and PM10: Please make the numbers 2.5 and 10 subscripts.
The “summary and conclusions” part is too give. It’s boring to read. Please do not include discussion there. You may move any discussion to the previous section. Make this section integrated and compact.
Author Response
Please, see the attachment

Reviewer 2 Report
Journal: IJERPH
Manuscript ID: ijerph-1375781
Title: Impact of the COVID-19 lockdown in a European regional monitoring network (Spain): are we free from pollution episodes?
Authors: Gotzon Gangoiti, Maite De Blas , María Carmen Gómez , Ana Rodríguez , Eduardo Torre-Pascual , Estíbaliz García-Ruiz , Estíbaliz Sáez de Cámara , Iñaki Zuazo , José Antonio García , Verónica Valdenebro
This manuscript shows the impacts of the COVID-19 lockdown on air quality in the Basque county in Northern Spain during March –June 2021. The authors looked at trends of six pollutant: NO, NO2, O3, SO2, PM2.5 and PM10 prior, during and post the lockdown period. The manuscript is well-written however there are minor grammar mistakes that need to be corrected throughout the manuscript. The manuscript thoroughly introduced the BC region, meteorology, lockdown, etc. However, the results and discussion section could be organized a little bit better and the authors can still use the terms rural or urban when talking about the different cities. Otherwise the readers will have to go every time to table 1 to check what are the abbreviation and if the city is rural or urban. I think the authors need to do few minor edits and answer some of the questions below before the manuscript is accepted
General comment:
Looking at the data in Table 6 and the results and discussion section, it does not seem that there were significant decrease in PM 10, PM 2.5 and maybe ozone. The authors suggested that this is due to a combination of an African dust storm/forest fire that impacted the region and might have affected the results. Wouldn’t it be better to remove those episodes during the same year comparisons and comparison to other years? The paper objective was to compare the effect of lockdown on air quality and having those anomaly weeks are biasing the conclusion and affecting the outcome of the paper. I think the authors need to do a comparison without those days for the period in 2020 (w-4 to w14 ) and between 2020 and 2015-2019 data in order to get a better idea about the impact of lockdown on air quality in BC.
Other comments:
Page 2, line 76 “in the air pollution” needs to be “on …”
Page 5, line 138 : This organism?? Do you mean organization? I think the word organism is wrong here
Page 5 lines 140-149, the sentence is so long and complex maybe the authors can revise it and break it down to smaller sentences.
Page 18 lines 622-640, are there other sources in DS that might have impacted the ozone chemistry? Is DS located in a valley compared to the other urban areas?
Page 22, lines 700-709, Were there any fires or African dust episodes observed in 2015-2019 to undergo this comparison?
Page 23, lines 738 -740 the sentence is not very clear
Author Response
Please, see the attachment

Reviewer 3 Report
General comments
This article presents the impact of the closures due to the pandemic on the air quality of criteria pollutants in three cities and rural areas in north-eastern Spain. A research question is presented in the title that is neither addressed nor answered throughout the article. In general, it is a very long manuscript and difficult to read, mainly due to the disorder and redundancy of some sections. Some sections and figures are unnecessary, so it should be reordered and rewritten before being reviewed again. Below I will summarize my main concerns regarding the different sections of the article.
1.- Only section 3.2 contributes to show the impact of the lockdown due to the pandemic on air quality. Methodology used in the other sections is either not properly justified (sections 3.3 and 3.4) or is not focused on showing the possible effect of other variables on the relative changes observed in concentrations (section 3.1). The question postured in the title of the article is not developed or answered at all in any section of the article. In relation to this point, summary and conclusions section only refers in depth to section 3.2 and mentions section 3.4 superficially (lines 858-868) making it clear that sections 3.1, 3.3 and 3.4 contribute nothing or very little to the research aim set out in the article.
2.- The introduction requires a more detailed review of the findings in other cities in Spain, Europe and the world in relation to the effect of lockdowns on air quality. There is extensive recent literature on the matter to give the reader an idea of ​​the effects of reducing emissions on air quality in large cities around Spain and the world.
3.- In section 2.1 must include general climatological information of the study area, (temperatures, relative humidity, annual rainfall, seasonality, etc.). Section 2.3 is very extense and is part of the results of the article. This section should just include the sources of information and methods used to estimate the variations in emissions between 2019 and 2020.
4.- Section 3.3 does not substantially provide any additional analysis or discussion to that indicated in section 3.2, so I suggest remove it, especially figures 5 and 6 that only confirm that there is no effect of restrictions on the O3 and PM10 levels, so they could be taken to supplementary material or removed. The article has 9 figures which is a too much, eliminating section 3.3 would be left with 6 figures.
5.- Section 3.4 makes a very detailed description of the events that could explain the air pollution episodes in 2020. However, the comparison between the concentrations of 8-h (for O3) or 24-h (for PM's) between 2015-2019 vs 2020 does not make sense. What relationship could exist between the average concentration levels of a day or in a particular week of the year 2020 with the average levels in the same days or week of the 2015-2019 period? The "anomalies" indicated are simply air pollution episodes driven by synoptic and/or local meteorological conditions that have no relationship with the calendar days. This methodology to describe the effect of lockdowns measures on air quality is not justified in the text or any article cited by the authors. In fact, the shaded area (interannual variability 2015-1019) of figures 6, 7, 8 and 9 remains covering the red line (year 2020) for much of the time series, so it could be argued that the variations observed in 2020 are typical of the interannual variability in the study area.
6.- The use of “a cluster-station data” for each city must be justified based on the degree of similarity between the concentration levels of the different stations within each city. The authors point out that the cities under study are relatively large, so statistically significant differences could be expected between the concentrations recorded at the stations within each city. Averaging the concentrations for the entire city could hide concentration peaks and its variability. Due the authors note that the article presents an evaluation of the concentration series at a daily / hourly resolution focused on the occurrence of episodes during the period, this point could be specially important. The divergence coefficient (DOI: 10.1080/02786821003749509) could be useful or a basic non-parametric statistical analysis to evaluate the statistical significance of the differences between the stations in each city. If the concentrations of any pollutant are statistically different between stations, the use of averages to calculate a cluster-station data for each city would seriously compromise the results presented.
7.- In order to improve the presentation of the results, the methodology or results sections should be reordered. The methodology ends with the meteorological analysis (section 2.4) while the results begin with the meteorological analysis (section 3.1).
Specific Comments
Line 15 and 77: Implantation? Find a better word that describes “intensive industrial activity”?.
Line 142: it is indicated that 13, 5 and 4 stations will be used for BI, VG and DS, respectively, however in table 1 and in figure 1c 5, 4 and 4 stations are mentioned for each city.
Line 145: Before doing a cluster-station data for each city, the authors should demonstrate that there are no statistically significant differences between the pollutant measurements between the stations within each city.
Table 2: The term “relaxation” is weird and could lead to misinterpretations. Maybe “release” or “opening”? Correct throughout the whole manuscript.
Line 165-180: The description of the restrictions is very general. The difference between stages 0, 1, 2 and 3 of the "relaxation" level, for example, could be better explained. The same applies to the other levels indicated. This would help to understand the effect that restrictions would have on the emissions patterns of the pollutants.
Table 4: There are contributions that add up to 101%, possibly due to the use of significant figures. Please check.
Table 5: The increase in power plants (+ 57%) should be in red.
Line 293. A character between 35% and 37% was lost.
Lines 281-297: This analysis is not understood if there is no previous description of the wind regime in BC that could influence transboundary pollution in the study area (this description could be included in section 2.1). It is not clear why the authors refer to the reduction of emissions in Italy, Austria or Portugal? None of these countries borders BC and it is not understood how they could influence the concentration levels of pollutants in BC.
Lines 321-325: This has already been said in the methodology and is not part of the results section.
Line 325-330: This information should be in the methodology.
Line 330-337: This entire description is unnecessary. The results section could simply begin with section 3.1.
Line 344: “vertical venting”? Replace by “vertical ventilation”. Correct the term "venting" throughout the text.
Line 339-359: These are not results. Bring all this information to the description of the study area and climatology (section 2.1)
Figure 3: The quality of this figure needs to be improved, there are unreadable numbers and the scale of the color palette is not understood (very small numbers).
Lines 360-398: The meteorological analysis presented here is poor and not well oriented. This analysis should be focused on answering whether the ventilation conditions could partially explain or not the relative changes on pollutant concentrations levels between 2020 and previous years. I suggest leaving this section for the end of the article and analyzing in relation to the relative changes observed in section 3.2.
Table 6: the high SD observed confirm that there are significant variations between the different sites within each city.
Lines 416-448: This section is very long and even unnecessary considering the purpose of the article, which is developed in the following sections (3.2.2 onwards).
Line 427: VI *? It should be VG *
Figure 4: The horizontal solid line in sections (c) and (d) of this figure does not make any sense (It’s an annual WHO guideline). The numbers and texts on the scales of these figures c and d are very small they can hardly be read.
Line 593-600: The comparison described here between the weekly averages and the WHO annual standard does not make any sense. The annual guidelines serve as a basis of comparison for establishing air quality regards to annual averages and not should be used to compare with averages of shorter duration (monthly or weekly). Please remove.
Line 636-648: This analysis only indicates that there is no weekend effect in the cities studied. There is also no variation in concentrations between the period 2015-2019 and the year 2020. In fact, in many periods the variability of the period 2015-2019 is covering the period 2020.
Line 692: The spacing increased.
Figures 8 and 9. Comparison of 24-h concentrations with the WHO annual guidelines does not make sense. Remove and only keep the daily guideline (24-h).
Lines 858-868: Again, the comparison between annual guidelines and 24-hour concentration levels does not apply here.
Author Response
Please, see the attachment

Round 2
Reviewer 2 Report
The authors answered most of my comments in the revised manuscript